# *Flux4D*: Flow-based Unsupervised 4D Reconstruction

**Jingkang Wang**[1,2*]   **Henry Che**[1,3*†]   **Yun Chen**[1,2*]   **Ze Yang**[1,2]
**Lily Goli**[1,2†]   **Sivabalan Manivasagam**[1,2]   **Raquel Urtasun**[1,2]

Waabi[1]    University of Toronto[2]    UIUC[3]

https://waabi.ai/flux4d

## Abstract

Reconstructing large-scale dynamic scenes from visual observations is a fundamental challenge in computer vision. While recent differentiable rendering methods such as NeRF and 3DGS have achieved impressive photorealistic reconstruction, they suffer from scalability limitations and require annotations to decouple moving actors from the static scene, such as in autonomous driving scenarios. Existing self-supervised methods attempt to eliminate explicit annotations by leveraging motion cues and geometric priors, yet they remain constrained by per-scene optimization and sensitivity to hyperparameter tuning. In this paper, we introduce *Flux4D*, a simple and scalable framework for 4D reconstruction of large-scale dynamic driving scenes. *Flux4D* directly predicts 3D Gaussians and their motion dynamics to reconstruct sensor observations in a fully unsupervised manner. By adopting only photometric losses and enforcing an "as static as possible" regularization, *Flux4D* learns to decompose dynamic elements directly from raw data without requiring pre-trained supervised models or foundational priors simply by training across many scenes. Our approach enables efficient reconstruction of dynamic scenes within seconds, scales effectively to large datasets, and generalizes well to unseen environments, including rare and unknown objects. Experiments on outdoor driving datasets show *Flux4D* significantly outperforms existing methods in scalability, generalization, and reconstruction quality.

## 1   Introduction

Reconstructing the 4D physical world from visual observations captured in the wild is a key goal in computer vision, with applications in virtual reality and robotics, including autonomous driving. High-quality reconstructions provide the foundation for scalable simulation environments that enable safer and more efficient autonomy development. Unlike artist-created environments, environments built automatically with data collected by sensor-equipped vehicles are more realistic, are more cost-efficient, and capture the diversity of the real world.

Advances in differentiable rendering approaches such as Neural Radiance Field (NeRF) [26] and 3D Gaussian Splatting (3DGS) [17] have enabled high-quality reconstruction of dynamic scenes [53, 50, 62, 39, 18]. These methods decompose scenes into a static background and a set of dynamic actors using human annotations such as 3D tracklets or dynamic masks, and then perform rendering on the composed representation, optimizing to reconstruct the input observations. While they achieve impressive visual fidelity, their reliance on manual annotations to decompose static and dynamic elements increases costs and time, preventing these methods from scaling to large sets of unlabelled data. Some approaches leverage pre-trained perception models to generate annotations automatically, but this can cause artifacts when the model predictions are noisy or incorrect, which can be difficult to recover from during reconstruction. Moreover, these methods typically require hours to reconstruct

---

[*]Equal contributions.

[†]Work done while a research intern at Waabi.

39th Conference on Neural Information Processing Systems (NeurIPS 2025).

each scene on consumer GPUs. These two main issues, expensive annotation costs and slow per-scene optimization, limit the scalability of these methods.

Recent works have explored self-supervised approaches to eliminate the reliance on human annotations and learn the decomposition of static and dynamic actors directly from data. This is a challenging task due to the ambiguity of actor motion over time, coupled with spatial geometry and appearance variations. One strategy attempts to improve the decomposition by incorporating additional regularization terms such as geometric constraints [31] or cycle consistency [52], or performing multi-stage training [16]. Another strategy is to leverage foundation models for additional semantic features or priors [31, 7, 52]. However, the resulting complex models can be sensitive to hyperparameters, slow to train, and unable to generalize to new scenes. Moreover, they often have poor decomposition results, and struggle to render novel views, limiting their usability.

As an alternative to costly per-scene optimization, generalizable approaches [3, 42, 2, 5, 13, 44, 59] use feed-forward neural networks to predict scene representations directly from observations, enabling efficient reconstruction within seconds. However, these approaches are designed for small-scale environments, can only process a few low-resolution images (typically 1-4 views with resolutions below 512px), and primarily focus on static scenes [2, 5] or only dynamic objects [33]. When handling large scenes with many dynamic elements, they rely on costly annotations [6, 34], limiting their scalability. Most recently, DrivingRecon [25] and STORM [51] propose feed-forward, self-supervised approaches for driving scenes. While promising, these methods focus on the sparse reconstruction setting and can only handle a small number ($\leq 12$) of low-resolution ($\leq 360$px) input views before reaching compute limits, and still depend on pre-trained vision models for semantic guidance, constraining their fidelity, scalability and applicability to downstream simulation.

In this paper, we propose *Flux4D*, an *unsupervised* and *generalizable* reconstruction approach that enables accurate and efficient 4D driving scene reconstruction at scale. Without any annotations, *Flux4D* predicts 3D Gaussians along with motion parameters directly in 3D space from multi-sensor observations within seconds, enabling efficient scene reconstruction. Our reconstruction paradigm is illustrated in Fig. 1. *Flux4D* uses a remarkably minimalist design that employs only photometric losses and a simple static-preference prior, without requiring complex regularization schemes or external supervision to learn the motion that prior works leverage. We find that the key ingredient for *Flux4D* to accurately recover geometry, appearance, and motion flow comes from learning across a diverse range of scenes. Moreover, *Flux4D*'s use of LiDAR data, commonly available in the autonomous driving domain, enable handling of a large number ($\geq 60$) of high-resolution (1080px) input multi-view images, achieving high-fidelity reconstruction and scalable simulation. Our 3D design yields a compact and geometrically consistent representation across views, improving efficiency, enabling explicit multi-view flow reasoning and reducing appearance-motion ambiguity.

Experiments on outdoor driving datasets PandaSet [48] and WOD [36] demonstrate that *Flux4D* achieves better scene decomposition and novel view synthesis than previous state-of-the-art annotation-free reconstruction methods, and is competitive with per-scene optimization methods that use human annotations. We also show that *Flux4D* can be trained to predict sensor observations in future frames, akin to next-token prediction, but applied to dynamic 3D scenes. Finally, we showcase using *Flux4D*'s reconstruction for controllable camera simulation via scene editing and novel view rendering at high resolution ($\geq 1080$px). *Flux4D* highlights the power of unsupervised learning for 4D scene reconstruction, enabling efficient scaling to vast unlabeled datasets.

## 2 Related Work

**Optimization-based 4D reconstruction:**   Inspired by differentiable rendering [26, 17], recent approaches use deformation fields [32, 30, 56, 46] to model dynamic scenes but still struggle with real-world complexity due to overparameterization and poor static-dynamic decomposition. While some methods address this by using human annotations (3D tracklets, semantic models) to explicitly separate static and dynamic elements [29, 54, 40, 50, 9, 12], they remain limited by annotation quality and availability. Self-supervised alternatives using motion cues and physics-informed priors [47, 52, 7, 16, 31] reduce dependence on annotations but typically require complex regularization schemes and expensive per-scene optimization. In contrast, our approach reconstructs dynamic 4D scenes without explicit supervision or per-scene optimization, achieving scalable reconstruction through simple photometric losses with minimal regularization.

**Generalizable reconstruction:**   Generalizable methods infer scene representations directly from observations without per-scene optimization [3, 42, 2, 5, 13, 44, 59], leveraging large training datasets

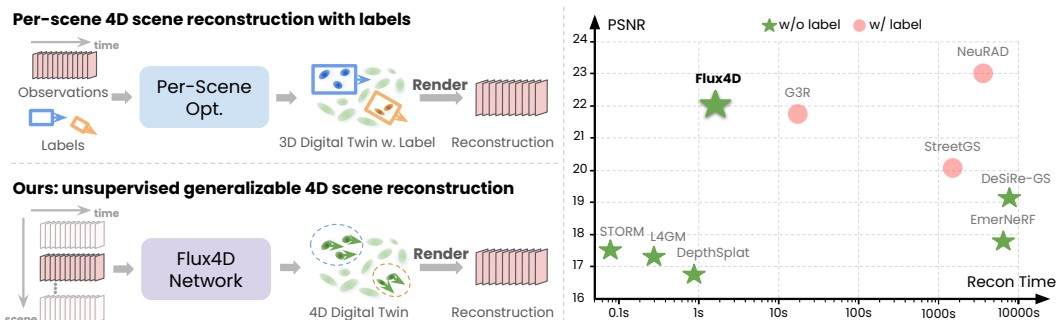

Figure 1: **Flux4D is a simple and scalable framework for unsupervised 4D reconstruction. Left:** Paradigms for 4D reconstruction. **Right:** realism-speed comparisons with existing works.

to improve reconstruction quality in novel environments. However, existing approaches primarily target static scenes, struggling with dynamic environments due to computational constraints and dependence on sparse, low-resolution inputs. Recent advances attempt to overcome these limitations using efficient architectures [63] or iterative refinement [6], but still rely on 3D annotations. In contrast, *Flux4D* generalizes to unseen dynamic scenes by predicting 3D Gaussians with their motion directly from raw observations without external supervision.

**Unsupervised world models:** Our work relates to recent advances in unsupervised world models, which learn predictive representations of environments without explicit supervision. These approaches typically tokenize visual data into discrete or continuous representations [14, 11, 43, 61, 27] processed by autoregressive or diffusion-based models to predict future states. While demonstrating impressive visual quality, such methods generally lack interpretable 3D structure, limiting precise control over generated content. Existing solutions often produce lower-resolution outputs with reduced temporal consistency, are typically restricted to single modalities (*e.g.*, camera [14, 11, 22] or LiDAR [60, 55, 1]), and require substantial computational resources. While our primary focus is reconstruction, *Flux4D*'s ability to simultaneously model motion dynamics and predict future frames shares conceptual similarities with world models. Unlike these approaches, *Flux4D* uses explicit 3D representation, providing 3D interpretability, controllability and spatiotemporal consistency.

**Unsupervised generalizable reconstruction:** Most recently, DrivingRecon [25] and STORM [51] explore unsupervised generalizable 4D reconstruction for driving scenes, using feed-forward networks to predict the velocities of 3D Gaussians. Despite impressive performance, they can process only sparse (3-4), low-resolution ($\leq 256 \times 512$) frames with substantial computational requirements and rely on pre-trained vision models (DeepLabv3+ [4], SAM [20], ViT-Adapter [8]) for additional supervision, limiting their scalability and applicability. *Flux4D* achieves better performance with a simpler and more scalable approach, and through our novel incorporation of LiDAR to initialize the scene, can handle full HD images with denser views ($> 60$) while being computationally efficient. Please see supp. for more discussions.

## 3 Scalable 4D Reconstruction with *Flux4D*

Given a sequence of camera and LiDAR data captured by a robot sensor platform, we aim to reconstruct the underlying 4D scene representation that disentangles static and dynamic entities and supports high-quality rendering at novel viewpoints. Such a representation can enable future prediction and counterfactual simulation. To achieve scalable 4D scene reconstruction, our method should be unsupervised, meaning it uses no annotations, and fast, running in seconds. Towards this goal, we propose *Flux4D*, an unsupervised and generalizable approach that learns to reconstruct 4D scenes via three simple steps (Fig. 2). We first lift the sensor observations at each timestep to a set of initial 3D Gaussians. We then feed the initial representation to a network to predict 3D flow and refined attributes for each 3D Gaussian. Finally, we supervise the network solely through reconstruction and static-preference losses.

### 3.1 Scene Representation

Our approach takes a set of posed camera images $\mathcal{I} = \{\mathbf{I}_k\}_{1 \leq k \leq K}$ and LiDAR point clouds $\mathcal{P} = \{\mathbf{P}_k\}_{1 \leq k \leq K}$ captured over time by a moving platform and outputs a scene representation

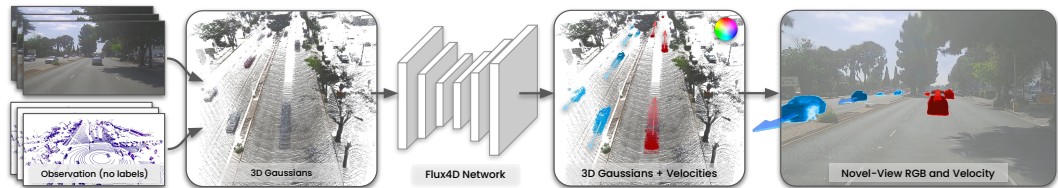

Figure 2: **Model overview.** *Flux4D* reconstructs 4D world by predicting 3D Gaussians with velocities given unlabelled sensor observations, and trained with the photometric reconstruction objective. The resultant model can be used for RGB and flow synthesis from novel views.

with geometry, appearance, and 3D flow. We represent the scene using a set of 3D Gaussians $\mathcal{G} = \{\mathbf{g}_i\}_{1 \leq i \leq M}$. Each Gaussian point $g_i$ is parameterized by its center position $\mathbf{p}_i$ ($\mathbb{R}^3$), scale ($\mathbb{R}^3$), orientation ($\mathbb{R}^4$), color ($\mathbb{R}^3$) and opacity ($\mathbb{R}^1$) [17]. Additionally, we augment each Gaussian with a learnable instantaneous velocity $\mathbf{v}_i \in \mathbb{R}^3$ and a fixed capture time $t_i$. We denote the sets of velocities and timestamps for all Gaussians as $\mathcal{V} = \{\mathbf{v}_i\}_{1 \leq i \leq M}$ and $\mathcal{T} = \{t_i\}_{1 \leq i \leq M}$.

**Initialization:** We initialize Gaussian positions from LiDAR points $\mathbf{P}_k$ from each source frame in the sequence, set scales based on the average distance to nearby points, and assign colors by projecting these points onto the corresponding camera image $\mathbf{I}_k$. Each Gaussian's timestamp $t_i$ is assigned the capture time of its source LiDAR frame, and velocities are initialized to zero. We aggregate source frame Gaussians to create $\mathcal{G}_{\text{init}}$.

### 3.2 Predicting Flow and Rendering

Inspired by recent advances in 4D reconstruction [47, 52, 31, 58, 25, 51], we propose to learn a time-dependent velocity field to model the dynamics of driving scenes. Given the initial velocity-augmented Gaussians $\mathcal{G}_{\text{init}}$, we leverage a neural reconstruction function $f_\theta$ that outputs the refined Gaussian parameters $\mathcal{G}$ and the predicted velocities $\mathcal{V}$:

$$\mathcal{G}, \mathcal{V} = f_\theta(\mathcal{G}_{\text{init}}, \mathcal{T}). \tag{1}$$

With the predicted velocities $\mathcal{V}$, each Gaussian can be propagated from its initial timestep $t_i$ to any target timestep $t'$ using a linear motion model:

$$\mathbf{p}_i^{t'} = \mathbf{p}_i^{t_i} + \mathbf{v}_i \cdot (t' - t_i), \tag{2}$$

where $\mathbf{p}_i^{t'}$ is the Gaussian position at time $t'$, $\mathbf{v}_i$ and $t_i$ are its velocity and capture time. This formulation enables continuous, temporally consistent reconstruction under a constant velocity assumption. We find this simple motion model can already achieve reasonable performance when reconstructing outdoor driving scenes with short time horizons ($\sim 1s$), an observation aligned with existing works [31, 25, 21, 51]. Moreover, we investigate higher-order polynomial motion models, as discussed in Sec. 3.4 and Table 7.

### 3.3 Unsupervised Learning of Dynamics

We now describe how the method learns to disentangle the scene dynamics. The network $f_\theta$ is trained in a fully self-supervised manner, without requiring explicit 3D annotations. Given the predicted Gaussians $\mathcal{G}$, we move the Gaussians to target time $t'$ using Eqn. (2), render the scene using differentiable rasterization [17] to generate color and depth images, and compare them against the real sensor observations $\mathcal{I}$ and $\mathcal{P}$. To prevent unnecessary motion and encourage stability, we introduce an "*as static as possible*" regularization. The total loss $\mathcal{L}$ is defined as:

$$\mathcal{L} = \mathcal{L}_{\text{recon}} + \lambda_{\text{vel}}\mathcal{L}_{\text{vel}}, \tag{3}$$

where $\mathcal{L}_{\text{recon}}$ represents the reconstruction loss, consisting of $L_1$ and structural similarity losses w.r.t the images, and an $L_1$ depth loss in the image plane compared to the projected LiDAR, and $\mathcal{L}_{\text{vel}}$ serves as a velocity regularization term that minimizes motion magnitudes:

$$\mathcal{L}_{\text{recon}} = \lambda_{\text{rgb}}\mathcal{L}_{\text{rgb}} + \lambda_{\text{SSIM}}\mathcal{L}_{\text{SSIM}} + \lambda_{\text{depth}}\mathcal{L}_{\text{depth}}, \tag{4}$$

Table 1: **Comparison to SoTA unsupervised methods on novel view synthesis.** We evaluate photorealism, geometry, and speed metrics against per-scene optimization methods and generalizable methods. $^\dagger$ denotes the need for pre-trained vision models. *Flux4D* surpasses unsupervised and achieves competitive performance with supervised methods (top block), without requiring 3D labels.

| Methods | Unsup. | Gen. | Dynamic-only | | | | Full image | | | | Recon speed |
| | | | PSNR↑ | SSIM↑ | $D_{RMSE}$ ↓ | $V_{RMSE}$ ↓ | PSNR↑ | SSIM↑ | $D_{RMSE}$ ↓ | $V_{RMSE}$ ↓ | Time↓ |
|---|---|---|---|---|---|---|---|---|---|---|---|
| *Recon. with labels (reference)* | | | | | | | | | | | |
| NeuRAD [39] | ✗ | ✗ | 23.01 | 0.734 | 1.98 | – | 24.61 | 0.685 | 2.30 | – | ∼60min |
| StreetGS [50] | ✗ | ✗ | 20.06 | 0.605 | 1.02 | – | 23.38 | 0.680 | 0.84 | – | ∼28min |
| G3R [6] | ✗ | ✓ | 21.85 | 0.670 | 2.33 | – | 24.35 | 0.686 | 1.96 | – | 17s |
| *Unsupervised recon.* | | | | | | | | | | | |
| EmerNeRF$^\dagger$ [52] | ✓ | ✗ | 17.79 | 0.411 | 6.09 | 0.318 | 22.80 | 0.624 | 4.24 | 0.432 | ∼100min |
| DeSiRe-GS$^\dagger$ [31] | ✓ | ✗ | 19.08 | 0.477 | 3.36 | 0.297 | 22.25 | 0.608 | 24.89 | 0.322 | ∼120min |
| DepthSplat* [49] | ✓ | ✓ | 16.87 | 0.425 | 6.18 | – | 21.40 | 0.595 | 2.73 | – | 0.87s |
| L4GM [33] | ✓ | ✓ | 17.36 | 0.343 | – | – | 19.38 | 0.465 | – | – | 0.32s |
| STORM [51] | ✓ | ✓ | 17.65 | 0.367 | 5.24 | 0.203 | 20.79 | 0.508 | 4.80 | 0.238 | **0.07s** |
| *Flux4D* (Ours) | ✓ | ✓ | **21.99** | **0.662** | **1.63** | **0.157** | **23.84** | **0.675** | **1.07** | **0.182** | 3.9s |

$$\mathcal{L}_{\text{vel}} = \frac{1}{M} \sum_i \|\mathbf{v}_i\|_2. \tag{5}$$

We train $f_\theta$ across a diverse set of scenes. Notably, we find that training across many scenes enables the network to *automatically* decompose static and dynamic components in urban scenes without requiring the complex regularizations used in prior per-scene optimization techniques [47, 52, 7, 16, 31]. This highlights the effectiveness of data-driven priors as a powerful form of implicit regularization and the scalability of this simple framework.

### 3.4  Improving Realism and Flow

The aforementioned components form the core of our approach, termed *Flux4D*-base. *Flux4D*-base can already disentangle motion and render novel views with high quality. We further improve *Flux4D*-base through two enhancements that further recover more fine-grained appearance and refined flow, resulting in our final model, *Flux4D*.

**Iterative refinement:**  *Flux4D*-base recovers the overall scene appearance, but often lacks fine-grained details. We hypothesize that this limitation stems from the constrained capacity of a single-step feedforward network, and imperfect initialization due to occlusions. To mitigate this, we introduce an iterative refinement mechanism inspired by G3R [6], leveraging 3D gradients as feedback to enhance reconstruction quality. Specifically, after each forward pass and generation of rendered color and depth at the supervision views, we compute the 3D gradients of the Gaussians according to the loss function Eqn. (3), and provide the generated Gaussians and gradients as input to a network $f_\phi$ to further refine them. This process progressively corrects color inconsistencies and sharpens details within as few as two iterations. By incorporating iterative feedback, our method achieves higher-fidelity reconstruction, particularly in regions with complex appearance variations, while preserving the efficiency and scalability of *Flux4D*-base.

**Motion enhancement:**  *Flux4D*-base recovers the overall scene flow accurately (Table 7). We further introduce *polynomial motion* parameterizations to better model actor behaviors like acceleration, braking or turning. Please see supp. for more details and comparisons. Exploring more advanced velocity models [21] or implicit flow representations is an exciting direction for future work. To further improve the flow and appearance quality of dynamic actors, we modify the loss function to focus on dynamic regions. Specifically, we render the flow in the image plane and apply pixel-wise re-weighting to the photometric loss. This gives higher importance to faster-moving regions during training, which typically occupy fewer pixels and would contribute less to the overall loss.

## 4  Experiments

We evaluate *Flux4D* against the current state-of-the-art (SoTA) self-supervised scene reconstruction methods, including both per-scene optimization and generalizable approaches. We also report the performance of supervised methods that do require annotations to model dynamics as a reference. We perform experiments on multiple outdoor dynamic datasets and assess novel view appearance

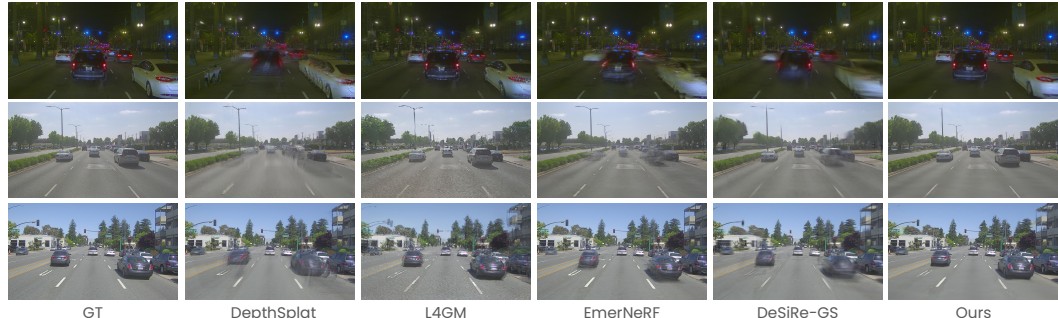

| GT | DepthSplat | L4GM | EmerNeRF | DeSiRe-GS | Ours |

Figure 3: **Qualitative results for NVS on PandaSet**. Rendered RGB images from novel views show that our method achieves better image quality across a variety of urban scenes, with crisper edges and sharper dynamic actors compared to baselines.

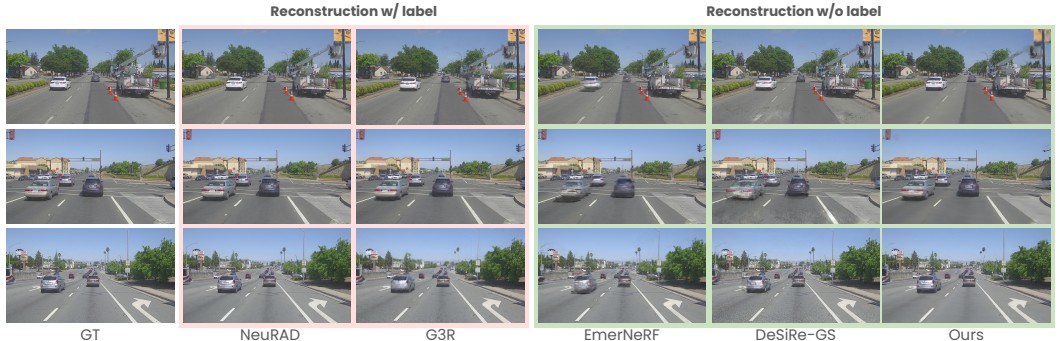

| Reconstruction w/ label | | | Reconstruction w/o label | | |

| GT | NeuRAD | G3R | EmerNeRF | DeSiRe-GS | Ours |

Figure 4: **NVS on longer-horizon logs.** Qualitative comparison shows that our method outperforms SoTA unsupervised baselines, by maintaining better estimation of actor movements in longer horizon. We shrink the gap in quality to supervised methods.

and depth, as well as recovered flow. We also ablate *Flux4D*'s design and show that *Flux4D* scales with more data. Finally, we demonstrate the controllability of our predicted scene representation for realistic camera simulation.

### 4.1 Experimental Details

**Experiment setup:** We conduct experiments on outdoor driving scenes from PandaSet [48] and Waymo Open Dataset (WOD) [36]. From PandaSet's 103 dynamic scenes (1080p cameras, 64-beam LiDARs, 10Hz), we select 10 diverse scenes for validation and use the rest for training. We use the front camera and 360° LiDAR, both collected at 10 Hz. To compare against existing feed-forward generalizable reconstruction methods that can only take a small number of frames as input, we report scene reconstruction results on short 1.5s windows within the validation sequences. Each method takes as input frames 0, 2, 4, 6, 8, 10, and is evaluated on frames 1, 3, 5, 7, 9 (*interpolation*) and 11-15 (*future prediction*). We sample a new snippet every 20 frames, yielding four non-overlapping evaluation snippets per log. We also evaluate against per-scene optimization methods over the full duration of the validation sequence (8 seconds) in the interpolation setting (every other frame is held out). For WOD evaluation, we follow the NVS setting in DrivingRecon [25], using the Waymo-NOTR subset with three front cameras, taking $\{t-2, t-1, t+1\}$ frames as input, and generating the interpolated frame at time $t$, where $t$ is every tenth frame in each sequence. Finally, we evaluate scene flow estimation perpformance on PandaSet and WOD (official validation set with 202 logs). As existing scene flow estimation methods cannot directly predict flows at novel timesteps, we evaluate scene flow on the input frames. We restrict evaluation to LiDAR points within the camera field of view (FoV) following [51].

**Baselines:** We compare against SoTA unsupervised scene reconstruction approaches: (1) *Self-supervised per-scene optimization:* EmerNeRF [52] and DeSiRe-GS [31], which reconstruct dynamic scenes using geometry priors, cycle consistency, and pre-trained vision models (FiT3D [57] and DI-

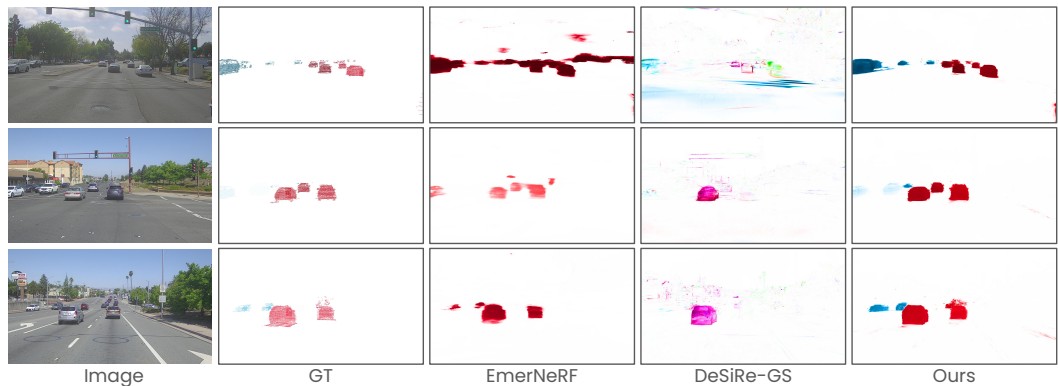

| Image | GT | EmerNeRF | DeSiRe-GS | Ours |

Figure 5: **Estimating motion flows.** We compare our estimated motion with prior unsupervised methods through rendered flow, showing accurate static region detection and sharper actor flow edges.

Table 2: **Full sequence reconstruction.** *Flux4D* outperforms unsupervised methods for 8-second reconstructions on dynamic regions and full image, closing the gap with supervised methods.

| Methods | Dynamic-only | | | | Full image | | | |
|---|---|---|---|---|---|---|---|---|
| | PSNR↑ | SSIM↑ | $D_{RMSE}$ ↓ | $V_{RMSE}$ ↓ | PSNR↑ | SSIM↑ | $D_{RMSE}$ ↓ | $V_{RMSE}$ ↓ |
| *Recon. with labels (reference)* | | | | | | | | |
| NeuRAD [39] | 22.99 | 0.719 | 1.71 | – | 24.99 | 0.679 | 2.29 | – |
| StreetGS [50] | 21.63 | 0.701 | 0.94 | – | 23.89 | 0.708 | 0.87 | – |
| G3R [6] | 20.60 | 0.573 | 2.16 | – | 23.15 | 0.636 | 2.01 | – |
| *Unsupervised recon.* | | | | | | | | |
| EmerNeRF† [52] | 18.65 | 0.437 | 4.48 | 0.478 | 23.42 | 0.627 | 3.09 | 0.975 |
| DeSiRe-GS† [31] | 19.76 | 0.544 | 4.08 | 0.312 | 22.91 | 0.659 | 4.07 | 0.395 |
| ***Flux4D* (Ours)** | **21.94** | **0.658** | **1.57** | **0.162** | **23.72** | **0.670** | **1.10** | **0.186** |

NOv2 [28]); (2) *Generalizable methods:* L4GM* [33], a 4D reconstruction model adapted to driving scenes using depth supervision; DepthSplat*, an extension of [49] that unprojects LiDAR points using estimated depth for 3D Gaussian prediction; DrivingRecon [25], which builds a 4D feed-forward model utilizing learned priors from pre-trained vision models (SAM [20] and DeepLab-v3 [4]); and STORM [51] which predicts per-pixel Gaussians and their motion in a feed-forward manner. For reference, we also include SoTA methods that use ground-truth 3D tracklets: StreetGS [50] and NeuRAD [39] (compositional 3DGS/NeRF), as well as G3R [6] (iterative refinement of compositional 3DGS). Apart from reconstruction methods, we also compare with representative scene flow estimation methods NSFP [23] and FastNSF [24] as a reference.

**Metrics:** We report standard metrics to measure the photorealism, geometric and motion accuracy using PSNR, SSIM, and depth RMSE ($V_{RMSE}$) and velocity RMSE ($V_{RMSE}$). Results are reported on both full images and dynamically moving regions for a comprehensive assessment. For scene-flow quality, we report EPE3D, $Acc_5$ and $Acc_{10}$ (fraction of points with error $\leq$ 5/10 cm), angular error in radians ($\theta_\epsilon$), three-way EPE [10]: background-static (BS), foreground-static (FS), and foreground-dynamic (FD), bucketed normalized EPE [19], and inference speed. On WOD, where semantic labels are coarse, we follow EulerFlow [41] and report bucketed normalized EPE for *Background (incl. Signs)*, *Vehicles*, *Pedestrians*, and *Cyclists* only.

***Flux4D* implementation details:** We adopt a 3D U-Net with sparse convolutions [37] for $f_\theta$. To handle unbounded scenes, we place random points on a spherical plane at a far distance to model sky and far-away regions. We also add random points within a 3D sphere following [50] to increase model robustness. Our model processes full-resolution images ($\geq 1920 \times 1080$) in all experiments and can be efficiently scaled to higher resolutions without significant overhead. Unless otherwise stated, all models are trained for 30,000 iterations on $4\times$ NVIDIA L40S (48G) GPUs, taking approximately 2 days. The reconstruction loss weights $\lambda_{rgb}, \lambda_{SSIM}, \lambda_{depth}$ are set as 0.8, 0.2 and 0.01 respectively. The velocity regularization weight $\lambda_{vel}$ is set as 5e-3.

Table 3: **NVS on WOD [36].** We achieve significant improvements over generalizable baselines.

| Methods | Full Image | | | Dynamic | | Static | |
|---|---|---|---|---|---|---|---|
| | PSNR | SSIM | LPIPS | PSNR | SSIM | PSNR | SSIM |
| LGM [38] | 17.49 | 0.47 | 0.33 | 17.79 | 0.49 | 15.37 | 0.39 |
| PixelSplat [2] | 18.24 | 0.56 | 0.30 | 18.63 | 0.58 | 16.96 | 0.44 |
| MVSplat [5] | 19.00 | 0.57 | 0.28 | 19.29 | 0.58 | 17.35 | 0.47 |
| L4GM [33] | 17.63 | 0.54 | 0.31 | 18.58 | 0.56 | 16.78 | 0.43 |
| DrivingRecon [25] | 20.63 | 0.61 | 0.21 | 20.97 | 0.62 | 19.70 | 0.51 |
| *Flux4D* | **26.62** | **0.82** | **0.18** | **26.86** | **0.83** | **26.09** | **0.80** |

Table 4: **Future prediction.** We surpass unsupervised and supervised methods.

| Methods | PSNR↑ | SSIM↑ | $D_{RMSE}$ ↓ | $V_{RMSE}$ ↓ |
|---|---|---|---|---|
| *Recon. with labels* | | | | |
| NeuRAD [39] | 21.52 | 0.557 | 3.03 | – |
| StreetGS [50] | 19.09 | 0.499 | 1.49 | – |
| G3R [50] | 21.13 | 0.570 | 2.09 | – |
| *Unsupervised recon.* | | | | |
| EmerNeRF† [52] | 19.64 | 0.516 | 5.00 | 0.346 |
| DeSiRe-GS† [31] | 18.86 | 0.513 | 26.07 | 0.325 |
| STORM [51] | 19.63 | 0.489 | 5.19 | 0.251 |
| *Flux4D* (Ours) | **21.81** | **0.598** | **1.42** | **0.193** |

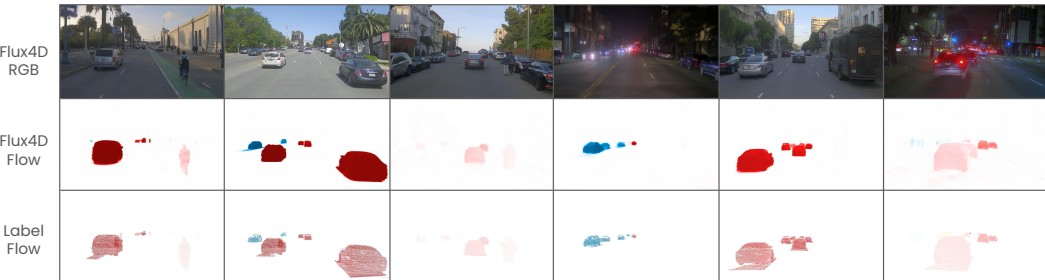

Figure 6: **High-fidelity flow and RGB reconstruction.** *Flux4D* not only provides photorealistic reconstruction of the dynamic scene but also estimates actors' motion flow with high precision.

## 4.2 Scalable 4D Reconstruction

**Novel view synthesis on PandaSet:** Table 1 and Fig. 3 compare *Flux4D* against SoTA unsupervised methods on 1s PandaSet snippets in the interpolation setting, with supervised approaches included for reference. Reconstruction speed is measured on a single RTX A5000 GPU (24GB). *Flux4D* achieves superior photorealism and geometric accuracy with fast reconstruction speed. We further evaluate our method on longer-horizon reconstruction of 8 second logs (Table 2 and Fig. 4), using iterative processing of 1s snippets. Our approach outperforms unsupervised per-scene optimization methods by a large margin on both 1s and 8s reconstruction tasks, without requiring pre-trained models or complex regularization. Our quantitative results as reported in these tables also indicate that *Flux4D* is competitive even against supervised approaches. Qualitatively, as shown in Fig. 3 and 4, *Flux4D* achieves high-fidelity camera rendering in both static and dynamic regions, while existing unsupervised approaches usually suffer from noticeable artifacts on dynamic actors due to inaccurate learned dynamics.

**Novel view synthesis on WOD:** We further compare *Flux4D* with SoTA generalizable methods on WOD in Table 3, where we follow the setup in [25]. The baseline results are from DrivingRecon [25] paper and we confirmed the setup and results with the authors to ensure accurate comparison. *Flux4D* surpasses DrivingRecon by +5.99 dB in PSNR and +0.21 in SSIM, demonstrating its effectiveness for unsupervised dynamic scene reconstruction. Please see supp. for qualitative comparisons.

**Flow estimation:** We compare the estimated motion flows of *Flux4D* with existing unsupervised per-scene optimization methods EmerNeRF [52] and DeSiRe-GS [31]. As shown in Table 1, 2 and Fig. 5, *Flux4D* significantly outperforms prior approaches, learning accurate motion direction and magnitude without any supervision. In contrast, existing methods struggle to learn consistent motion flows and fully decompose dynamic scenes, leading to inaccurate and incoherent motion predictions, limiting their applicability in downstream tasks.

**Scene flow evaluation:** While *Flux4D* primarily focuses on reconstruction and is not specifically designed for scene flow estimation, we further evaluate its performance on PandaSet compared with representative scene flow estimation methods using standard scene flow metrics in Table 5 and 6. Please see supp. for comparisons on WOD. Although not designed for scene flow estimation, Flux4D achieves superior performance across most scene flow metrics using only reconstruction-based supervision (RGB + depth). Notably, it outperforms other methods on smaller or less common object categories such as wheeled VRUs, other vehicles, and pedestrians, as shown in bucketed evaluations. These results highlight a promising path to unifying state-of-the-art scene flow estimation and reconstruction within a single framework.

Table 5: **Comparison with scene flow estimation methods.**

| Method | EPE3D ↓ | $Acc_5$ ↑ | $Acc_{10}$ ↑ | $\theta_\epsilon$ ↓ | EPE-BS ↓ | EPE-FS ↓ | EPE-FD ↓ | EPE-3way ↓ | Inference time ↓ |
|---|---|---|---|---|---|---|---|---|---|
| NSFP [23] | 0.183 | 0.558 | 0.713 | 0.510 | 0.106 | 0.103 | 0.573 | 0.227 | ∼5.57 s/frame |
| FastNSF [24] | 0.194 | 0.571 | 0.714 | 0.471 | 0.155 | 0.134 | 0.428 | 0.211 | ∼0.68 s/frame |
| STORM [51] | 0.120 | 0.757 | 0.782 | 0.489 | **0.009** | **0.098** | 0.536 | 0.201 | **∼0.01 s/frame** |
| *Flux4D* | **0.094** | **0.775** | **0.807** | **0.123** | 0.019 | 0.117 | **0.391** | **0.165** | ∼0.31 s/frame |

Table 6: **Bucketed scene flow error on PandaSet.** Normalized EPE3D (↓) per class, split into static (S) and dynamic (D) regions. Mean S/D are averages across all buckets. Abbrev.: BG = Background, CAR = Car, WVRU = Wheeled VRU, VEH = Other Vehicles, PED = Pedestrian.

| Method | BG-S↓ | CAR-S↓ | CAR-D↓ | WVRU-S↓ | WVRU-D↓ | VEH-S↓ | VEH-D↓ | PED-S↓ | PED-D↓ | Mean S↓ | Mean D↓ |
|---|---|---|---|---|---|---|---|---|---|---|---|
| NSFP [23] | 0.128 | 0.093 | 0.668 | 0.046 | 0.975 | 0.060 | 0.819 | 0.071 | 0.945 | 0.080 | 0.852 |
| FastNSF [24] | 0.196 | 0.153 | **0.581** | 0.043 | 0.960 | 0.075 | 0.701 | 0.041 | **0.894** | 0.102 | **0.784** |
| STORM [51] | **0.005** | 0.087 | 0.713 | **0.000** | 1.000 | 0.195 | 1.000 | 0.093 | 1.012 | 0.076 | 0.931 |
| *Flux4D* | 0.019 | **0.078** | 0.701 | 0.011 | **0.866** | **0.021** | **0.661** | **0.027** | 0.966 | **0.031** | 0.800 |

**Future prediction:** We evaluate *Flux4D*'s capability for future frame prediction beyond the observed frames. This challenging task requires precise motion estimation, temporal consistency, occlusion reasoning, and a comprehensive 4D scene understanding. As shown in Table 4, *Flux4D* outperforms existing unsupervised methods in both photometric accuracy and geometric consistency. Moreover, *Flux4D* even outperforms supervised approaches that rely on imperfect explicit annotations for extrapolation, demonstrating the robustness of our predicted scene representation and the effectiveness of unsupervised scene flow prediction. This highlights *Flux4D*'s ability to model scene dynamics, which is critical for world modeling, simulation, and scene understanding in autonomous systems. We report full-image metrics in Table 4 and report dynamic-only metrics in supp.

**Ablation:** Table 7 evaluates *Flux4D*'s key design components. Iterative refinement significantly enhances image quality and geometric accuracy metrics. Polynomial motion modeling improves motion prediction performance. Table 8 demonstrates that our static-preference prior is essential to learning accurate flow, and that velocity reweighting improves performance on the dynamic elements. Please refer to supp. for qualitative comparisons.

**LiDAR-free *Flux4D*:** We show that *Flux4D* can also operate in a LiDAR-free mode at inference similar to DrivingRecon [25] and STORM [51] by using off-the-shelf monocular depth estimation model [15]. As shown in Table 9, the flow estimation performance remains comparable, and in some cases, the visual realism improves in background regions (*e.g.*, buildings) due to the broader coverage provided by monocular depth, particularly in areas where LiDAR sparsity limits reconstruction quality. Combining both LiDAR and points lifted by monocular depth yields the best overall realism.

**Scaling analysis:** *Flux4D*'s effectiveness stems from multi-scene training, leveraging diverse driving data as implicit regularization. Unlike per-scene methods that require complex regularizations or pre-trained models, increasing the amount of training data naturally improves scene decomposition and motion estimation. Analysis on PandaSet and WOD shows consistent improvements in photometric accuracy and motion estimation as training data scale. This confirms unsupervised 4D reconstruction benefits significantly from diverse real-world scenarios, suggesting *Flux4D* can continue improving with additional data, making it promising for scalable scene reconstruction.

**Camera Simulation:** We showcase applying *Flux4D* for high-fidelity camera simulation in large-scale driving scenarios. *Flux4D* produces high-quality motion flows in diverse, large-scale dynamic scenes on PandaSet (Fig. 6), Argoverse 2 [45], and WOD (Fig. 7). This allows accurate scene decomposition across diverse environments which is critical for instance extraction and direct manipulation of dynamic elements (Fig. 9). Compared to existing self-supervised per-scene methods, *Flux4D* is better suited for interactive and controllable applications, as it reconstructs an editable representation that supports instance mask extraction, scene editing and object manipulation for various downstream tasks. In Fig. 9, we demonstrate *Flux4D*'s capability to render realistic images of the modified scene representation. Notably, our approach achieves this without requiring labels.

## 5 Limitations

Although *Flux4D* achieves SoTA 4D reconstruction without any annotations or pre-trained models, three key limitations remain: (1) flow estimation for highly dynamic actors with complex motion

Table 7: **Ablation study on *Flux4D* designs.**

| Methods | Dynamic-only | | | |
|---|---|---|---|---|
| | PSNR↑ | SSIM↑ | $D_{RMSE}$↓ | $V_{RMSE}$↓ |
| *Flux4D-base* | 18.89 | 0.472 | 1.98 | **0.165** |
| + iterative refine | 21.32 | 0.636 | 1.66 | 0.167 |
| + polynomial motion | **21.45** | **0.641** | **1.55** | 0.167 |

Table 8: **Ablation study on training strategy.**

| Methods | Dynamic-only | | | |
|---|---|---|---|---|
| | PSNR↑ | SSIM↑ | $D_{RMSE}$↓ | $V_{RMSE}$↓ |
| *Flux4D* | **21.99** | **0.662** | 1.63 | **0.157** |
| − vel. reweighting | 21.45 | 0.641 | 1.55 | 0.167 |
| − vel. regularization | 21.08 | 0.614 | **1.44** | 0.532 |

Table 9: **LiDAR-free *Flux4D* using off-the-shelf monocular depth estimation model [15].**

| Methods | Dynamic-only | | | | Full image | | | | Scene Flow |
|---|---|---|---|---|---|---|---|---|---|
| | PSNR↑ | SSIM↑ | $D_{RMSE}$↓ | $V_{RMSE}$↓ | PSNR↑ | SSIM↑ | $D_{RMSE}$↓ | $V_{RMSE}$↓ | EPE-3way↓ |
| *Flux4D* (monocular depth only) | 21.71 | 0.668 | **1.45** | 0.159 | 23.87 | 0.688 | 1.23 | 0.186 | 0.165 |
| *Flux4D* (LiDAR, Table 1) | **21.99** | 0.662 | 1.63 | **0.157** | 23.84 | 0.675 | **1.07** | **0.182** | 0.165 |
| *Flux4D* (LiDAR + monocular depth) | **21.99** | **0.682** | 1.52 | 0.158 | **24.55** | **0.726** | 1.11 | 0.184 | **0.161** |

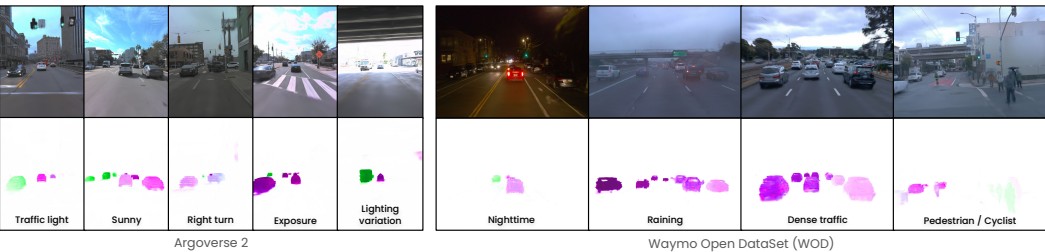

Figure 7: *Flux4D* reconstruction on Argoverse 2 and WOD.

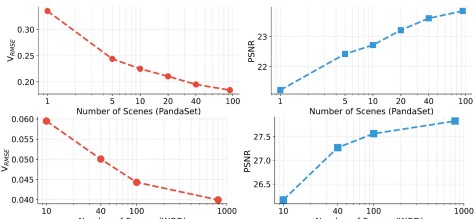

Figure 8: **Scaling analysis.** Increasing number of training scenes for *Flux4D* consistently improves performance.

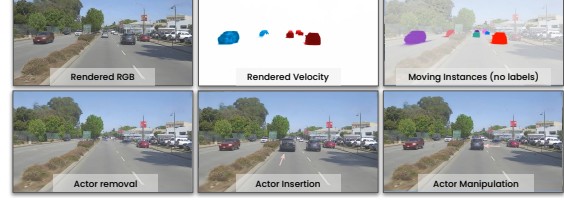

Figure 9: **Simulation applications.** Flux4D can be applied suc- cessfully to different camera simulation tasks, e.g., actor removal, insertion and manipulation.

patterns is challenging, which could be mitigated by leveraging larger and more diverse training data; (2) iterative approach for long-horizon reconstruction creates visible inconsistencies at transition points; and (3) the method assumes a simple pinhole camera model with clean LiDAR data, limiting applicability with rolling shutter cameras or noisy sensor inputs. Please see supp. for more examples. Future work will focus on scaling to larger datasets, developing a unified temporal representation for seamless long-term reconstruction, and improving robustness to real-world sensor imperfections. Furthermore, *Flux4D*'s explicit 3D representation offers interpretable structure for world models. Overall, we believe that our simple and scalable design serves as a foundation for the community to build upon, enabling further advancements in 4D reconstruction.

# 6 Conclusion

We present *Flux4D*, a scalable flow-based unsupervised framework for reconstructing large-scale dynamic scenes by directly predicting 3D Gaussians and their motion dynamics. By relying solely on photometric losses and enforcing an "as static as possible" regularization, *Flux4D* effectively decomposes dynamic elements without requiring any supervision, pre-trained models, or foundational priors. Our method enables fast reconstruction, scales efficiently to large datasets, and generalizes well to unseen environments. Extensive experiments on outdoor driving datasets demonstrate state-of-the-art performance in scalability, generalization, and reconstruction quality. We hope this work paves the way for efficient, unsupervised 4D scene reconstruction at scale.

## Acknowledgement

We sincerely thank the anonymous reviewers for their insightful suggestions especially on scene flow evaluation, paper framing, and additional experiments using monocular depth estimation models. We would like to thank Andrei Bârsan and Joyce Yang for their feedback on the early draft. We also thank the Waabi team for their valuable assistance and support.

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
