# OpenReview forum: "Flux4D: Flow-based Unsupervised 4D Reconstruction"
_NeurIPS.cc/2025/Conference — NeurIPS 2025 poster_

### Official Review · Reviewer_Ew2X · 2025-06-23

**Clarity:** 3
**Significance:** 2
**Originality:** 3
**Rating:** 4
**Confidence:** 2

**Summary:**

This paper presents an unsupervised training paradigm for NVS of dynamic scenes. At the core of this work, Lidar is used as a guiding modality for boot-strapping the GS initialization. Using Lidar information, a motion model is fitted as an additional regularization term in addition to a photometric loss composed of (rgb,ssim,depth). Comparisons are made with prior-works in the space showing improvement over unsupervised baselines and "closing the gap" to supervised approaches.

**Questions:**

Main point to address:
The applicability and methodology of this work as presented is good. It's more of a framing problem than a method problem. Imo with correct framing and emphasizing the need for lidar information this work can be better appreciated by the community. The presentation makes it hard to disentangle the scientific contribution w.r.t lidar based methods vs. methods that rely on foundation model priors.

**Ethical Concerns:**

["NO or VERY MINOR ethics concerns only"]

**Final Justification:**

In the rebuttal the authors show the method is robust to the quality of the geometric prior and can leverage mono-depth estimates as well while still performing well. I raised my score to a borderline accept.
The method is technically sound, however, the experiment section mixes methods that use different quality of depth inputs (model based, vs. lidar based). I encourage the authors for the camera ready to improve on this point.

**Limitations:**

See Questions

**Paper Formatting Concerns:**

No concerns

**Quality:**

2

**Strengths And Weaknesses:**

Strength:
- Good use of the lidar data for learning the motion model and GS initialization.
- Research emphasizes real-world applicability showing good runtime performance and good results on NVS for future frame prediction.

Weaknesses:
- The provided comparisons with previous works makes it hard to evaluate the scientific contributions of this work. While out-performing on the dry metrics, this work uses an additional modality which the others don't. It allows to build a motion model which isn't accesible to the other works.
- In light of this, I would place this work as an in-between the unsupervised and supervised works. I acknowledge that Lidar data can be collected in an unsupervised manner - in addition it's use is clever in extracting a form of velocity segmentation - other works do not have access to this information.

---

> ### Author Rebuttal · Authors · 2025-07-30
>
> Thank you for the thoughtful review and valuable feedback. We are glad that “the applicability and methodology of our work” are recognized as “good”. We now address the reviewer’s concerns regarding the framing of our method, particularly on our use of LiDAR.
>
> To clarify this, we (1) explain the dependency on LiDAR for baselines and provide additional experiments demonstrating that Flux4D can also operate in a LiDAR-free setting, by using off-the-shelf monocular depth estimation as an alternative, and (2) argue that using LiDAR should not be considered as a form of supervision in the context of self-driving applications, where LiDAR is commonly available and passively collected.
>
> **Q1: Dependence on LiDAR modality – Comparisons with previous works makes it hard to evaluate the scientific contributions**\
> **A1:** We acknowledge that our method leverages LiDAR to initialize our Gaussian scene representation and uses it for depth supervision. However, we would like to point out that all of our baselines (in Tab. 1, 2, 4) also have access to this data either in reconstruction or during training. Specifically, EmerNeRF, Desire-GS, G3R, StreetGS and NeuRAD all use LiDAR in their proposed reconstruction frameworks, two of which also predict scene flows (EmerNeRF, Desire-GS). For general feed-forward generalizable methods (e.g., DepthSplat, L4GM), we believe we have made considerable effort towards incorporating LiDAR in their frameworks (L219-222 and Supp. 105-123). While STORM and DrivingRecon use LiDAR during training as depth supervision, we acknowledge they do not use it at test time.
>
> To demonstrate Flux4D in a LiDAR-free setting at test time, similar to DrivingRecon and STORM, we replaced LiDAR point cloud initialization with metric depth estimated from foundational model Metric3D [1]. Specifically, we removed LiDAR points entirely and instead voxelized the projected monocular depth with a 5 cm resolution, retaining up to 2M points, and then trained the model following the same 3D processing pipeline.  Additionally, as suggested by Reviewer QsPn, we also include some representative scene flow metrics, namely Threeway-EPE (EPE-3way) [2] and Bucketed Normalized EPE (Mean S, Mean D) [3] to better evaluate the motion models.
>
> We observed that the flow estimation performance remains comparable with the original Flux4D, and in some cases, the visual realism improves in background regions (e.g., buildings) due to the broader coverage provided by monocular depth. Moreover, this LiDAR-free version of Flux4D performs significantly better than STORM. This demonstrates that our unsupervised learning of flow and appearance directly in 3D space with an approximate geometry initialization to be the key scientific contribution (as recognized by Reviewer QsPn), rather than the use of LiDAR.  We report the numbers below and will include the quantitative results, visualizations, and further analysis in the revision.
>
> | Method      |PSNR_D↑|SSIM_D↑|D-RSME_D↓|V-RMSE_D↓|PSNR↑|SSIM↑|D-RSME↓|V-RMSE↓|EPE-3way↓|Mean S↓|Mean D↓|
> |-------------|--------|--------|-----------|------------|------|------|-------|-------|------|-------|-------|
> |STORM        | 17.65  | 0.367  | 5.24      | 0.203      | 20.79| 0.508| 4.80  | 0.238 |0.201 | 0.076|0.931|
> |Flux4D (monocular depth only)  | 21.71  | 0.668  | **1.45**      | 0.159      | 23.87| 0.688| 1.23  | 0.186 |0.165|0.037|0.863
> |Flux4D (lidar, main paper)      | **21.99**  | 0.662  | 1.63      | **0.157**      | 23.84| 0.675| **1.07**  | **0.182** | 0.165|**0.031**|**0.800**|
> |Flux4D (lidar + monocular depth) | **21.99**  | **0.682**  | 1.52      | 0.158      | **24.55**| **0.726**| 1.11  | 0.184 | **0.161**|0.032|0.873
>
> **Q2: Unsupervised nature of Flux4D and framing**\
> **A2:** Thank you for raising this point. LiDAR is widely available and passively collected in real-world autonomous driving systems. Moreover, LiDAR in our framework is used purely as raw, sparse sensor input, without any human labels, semantic annotations, or ground-truth flow. The learning is driven entirely by photometric and geometric reconstruction losses, and Flux4D does not rely on supervised labels at any stage. This is consistent with definition and common practice in unsupervised learning, where additional modalities (e.g., lidar, stereo or depth) can be used as inputs and help define an energy or reconstruction objective [4, 5, 6]. Importantly, while LiDAR is sparse, Flux4D produces dense, photorealistic 3D reconstructions. We also note that existing unsupervised per-scene reconstruction methods for driving scenes (DesireGS, EmerNerf, UnIRe [7]) also require the usage of LiDAR during their reconstruction process.
>
> We also provide additional experiments in Q1/A1 demonstrating that Flux4D also performs well without LiDAR, using monocular depth estimated by an off-the-shelf model. We emphasize that this is fundamentally different from methods relying on foundation model priors as supervision, as such supervision can introduce inaccuracies and is not suitable for scalable unsupervised training. The key contribution of Flux4D lies in learning motion and scene dynamics directly from raw RGB and LiDAR inputs, without any annotated supervision.  We will clarify this framing in the revised manuscript and more explicitly discuss/emphasize the role of LiDAR within our unsupervised learning setup.
>
> We look forward to further discussions if you have any further questions or concerns.
>
> [1] Hu et al., Metric3d v2: A versatile monocular geometric foundation model for zero-shot metric depth and surface normal estimation, 2024. \
> [2] Chodosh et al., Re-Evaluating LiDAR Scene Flow for Autonomous Driving, 2024. \
> [3] Khatri et al., I Can't Believe It's Not Scene Flow!, 2024. \
> [4] Liu et al., Unsupervised Learning of Scene Flow Estimation Fusing with Local Rigidity, 2019. \
> [5] Li et al., Neural Scene Flow Prior, 2021. \
> [6] Vedder et al., Neural Eulerian Scene Flow Fields, 2025.  \
> [7] Mao et al., UnIRe: Unsupervised Instance Decomposition for Dynamic Urban Scene Reconstruction, 2025.

---

> ### Author Response · Authors · 2025-08-02
> **Thanks for the prompt reply!**
>
> We thank the reviewer for taking precious time in checking our responses. We hope our answers and additional results address your concerns well. Specifically,
>
> **Q1/A1:** We clarified baseline LiDAR dependencies and provided LiDAR-free Flux4D results. \
> **Q2/A2:** We clarified the unsupervised nature of Flux4D.
>
> Please let us know if you have any additional questions. We will be more than happy to clarify them. Any follow-up discussions are highly appreciated!

---

> > ### Comment · Reviewer_Ew2X · 2025-08-05
> >
> > I thank the authors for their additional results using off-the-shelf monocular depth.
> > This may strengthen the framing of the paper, as this makes a more general use of geometric priors - and not limited to LiDAR.
> > Additionally, given the results provided in Q1/A1, the quality of the geometric prior (i.e., SnR) plays a role on the final performance. In the paper this distinction is not clear making the results section difficult to parse.

---

> ### Author Response · Authors · 2025-08-05
>
> Thank you for the follow-up discussion! We are glad to hear that the additional results using off-the-shelf monocular depth help strengthen the framing of the paper. We appreciate the framing suggestion to emphasize that our approach leverages general geometric priors directly in 3D space to reduce ambiguity in reconstruction and flow estimation. We will revise our manuscript to better highlight this point.
>
> We also agree that the quality of the geometric prior influences the final performance: LiDAR + monocular depth slightly outperforms LiDAR-only, which in turn is slightly better than monocular depth alone. That said, our method remains highly robust even under imperfect Metric3D depth, thanks to training across diverse scenes and the use of iterative refinement. We will add discussions to better highlight this distinction and improve clarity.
>
> Please let us know if there are any additional concerns, questions, or suggestions, we would be happy to address them.

---

> > ### Author Response · Authors · 2025-08-07
> > **Discussion phase ending soon and thank you!**
> >
> > Thank you again for the thoughtful feedback and discussion. As the discussion phase is coming to a close (August 8, 11:59pm AoE), we wanted to check if there are any remaining questions or comments. We would be happy to address them promptly.
> >
> > Also, a gentle reminder to please fill in the final justification and update your score if appropriate. We sincerely appreciate your time and consideration throughout the review process!

---

### Official Review · Reviewer_Y9KP · 2025-07-02

**Clarity:** 3
**Significance:** 2
**Originality:** 2
**Rating:** 4
**Confidence:** 4

**Summary:**

The paper introduces a method for reconstructing vehicle driving scenes. By leveraging multi-sensor observations such as camera images and LiDAR point clouds, the method predicts 3D Gaussians and their motion dynamics. The optimization is guided by photometric losses and regularization terms, enabling the decomposition of dynamic elements from raw data without relying on annotations or pre-trained models.

**Questions:**

As stated in the Weaknesses section, it would be better to add more evaluations on various 4D scenes, not limited to vehicle driving scenarios. Additionally, even though the method is proposed for vehicle driving, its processing speed constraints limit practical applications.

**Ethical Concerns:**

["NO or VERY MINOR ethics concerns only"]

**Final Justification:**

The authors' detailed rebuttal addressed my concerns. Consequently, I have raised my score to "Borderline accept".

**Limitations:**

Yes

**Quality:**

3

**Strengths And Weaknesses:**

Strengths:
1. The paper is well-written and easy to follow.
2. The experiments and related analyses in the main paper and supplementary material are sufficient.

Weaknesses:
1. The experiments are conducted on outdoor driving scenes, which mainly involve sparse rigid motions and represent an easier part of general 4D scenes. It is inappropriate to use "4D" to name the method, or more evaluations on diverse 4D scenes—especially non-rigid motions where Eq. 2 is effective only within short time intervals are needed.
2. The network in Fig. 2 is claimed to be a generalizable module, but its generalization ability does not seem to be evaluated. Can it generalize to unseen 4D motions or just unseen vehicle velocities?
3. The inference speed of the method is slower than existing approaches, which restricts its application in vehicle driving scenarios.

---

> ### Author Rebuttal · Authors · 2025-07-30
>
> Thank you for the thoughtful review and valuable feedback. We are pleased that the paper was found to be “well‑written” and that the experiments and analysis are “sufficient.” We now address the concerns.
>
> **Q1: Scope beyond driving scenes: Inappropriate to use "4D" to name the method,evaluations on diverse 4D scenes**\
> **A1:** Thank you for pointing this out. As stated in the paper (Abstract: L8–10; Introduction: L59–60; Method: L118–121), Flux4D is a simple and scalable framework for 4D reconstruction of **large‑scale dynamic driving scenes**. We agree that driving scenes primarily involve sparse rigid motions. To better clarify this scope, we will consider renaming the submission to “FluxDrive: Unsupervised 4D Reconstruction for Driving Scenes.”
>
> We further note that our method handles short‑horizon non‑rigid motion (e.g., pedestrians and cyclists) on the Waymo Open Dataset and Argoverse 2, as shown in Fig. 7 thanks to the higher‑order polynomial motion models used in the final model (Sec. 3.4, Table 5, Appendix A.1). These motions and scenes are unseen during training, demonstrating the effectiveness and generalizability of our model on driving scenes. Regarding Eq. 2, it represents the motion model used in Flux4D-base, which can already achieve reasonable performance for outdoor driving scenes over short time horizons, as shown in our ablations. This observation aligns with existing works such as GVFi [1] and STORM [2].
>
> Finally, as acknowledged in the method and limitation sections, highly dynamic, complex motions remain challenging (L290–291). Exploring more advanced velocity models or implicit flow representations is an exciting direction for future work (L188–189).
>
> **Q2: Can it generalize to unseen 4D motions or just unseen vehicle velocities?**\
> **A2:** As discussed in Q1/A1, our method can generalize beyond vehicle velocities. In Fig. 7, we show that Flux4D can generalize to unseen non‑rigid motions (e.g., pedestrians and cyclists) on the Waymo Open Dataset and Argoverse 2. Moreover, as suggested by Reviewer QsPn, we conduct additional experiments to evaluate scene flow against test-time scene flow estimation methods (NSFP [3] and FastNSF[4]) using standard scene flow metrics, namely Threeway-EPE (EPE-3way) [5] and Bucketed Normalized EPE (Mean S, Mean D) [6].
>
> As shown in the table, Flux4D achieves competitive performance across most metrics. Notably, although our approach is designed for unsupervised reconstruction and simulation rather than scene flow estimation, it outperforms other methods on challenging cases, including smaller or less common object categories and non‑rigid actors such as wheeled VRUs, other vehicles, and pedestrians, as shown in bucketed evaluations. This highlights the generalizability and effectiveness of large-scale unsupervised training.
>
> ***Pandaset Table (1/2)***
> | Method |EPE3D↓|Acc@5↑|Acc@10↑|$\theta$ error↓|EPE-BS↓|EPE-FS↓|EPE-FD↓|EPE-3way↓|Inference↓|
> |--------|------|------|-------|---------|-------|-------|------|--------|------------|
> | NSFP   |0.183 |0.558 |0.713  |0.510    |0.106 |0.103 |0.573 |0.227   |~5.57 s/frame |
> | FastNSF|0.194 |0.571 |0.714  |0.471    |0.155 |0.134 |0.428 |0.211   |~0.68 s/frame |
> | STORM  |0.120 |0.757 |0.782  |0.489    |**0.009**|**0.098**|0.536 |0.201   |**~0.01 s/frame**|
> | Flux4D |**0.094**|**0.775**|**0.807**|**0.123**|0.019 |0.117 |**0.391**|**0.165**|~0.31 s/frame |
>
> ***Pandaset Table (2/2)***
> | Method |BG-S↓|CAR-S↓|CAR-D↓|WVRU-S↓|WVRU-D↓|VEH-S↓|VEH-D↓|PED-S↓|PED-D↓|Mean S↓|Mean D↓|
> |--------|-----|-----|-----|-----|-----|-----|-----|-----|-----|-----|-----|
> | NSFP   |0.128|0.093|0.668|0.046|0.975|0.060|0.819|0.071|0.945|0.080|0.852|
> | FastNSF|0.196|0.153|**0.581**|0.043|0.960|0.075|0.701|0.041|**0.894**|0.102|**0.784**|
> | STORM  |**0.005**|0.087|0.713|**0.000**|1.000|0.195|1.000|0.093|1.012|0.076|0.931|
> | Flux4D |0.019|**0.078**|0.701|0.011|**0.866**|**0.021**|**0.661**|**0.027**|0.966|**0.031**|0.800|
>
> **Legend:**  BG = Background, CAR = Car, WVRU = Wheeled VRU, VEH = Other Vehicles, PED = Pedestrian, S = Static, D = Dynamics
>
> **Q3: The inference speed of the method is slower than existing approaches, which restricts its application in vehicle driving scenarios.**\
> **A3:** Our method is designed for offline reconstruction and simulation, not real-time onboard perception. In this context, Flux4D offers significant advantages in both speed and reconstruction quality over state of the art, unveiling the possibility for scalable self-driving simulation.
>
> First, Flux4D largely bridges the gap compared to state-of-the-art supervised self-driving reconstruction methods (as recognized by Reviewers QsPn, tD1B, Ew2X). It is over **10× faster** than generalizable approaches like G3R while achieving higher-quality results. Compared to per-scene optimization methods such as StreetGS and NeuRAD, Flux4D offers up to **500×** speedup without requiring any labels, making it significantly more scalable for large-scale simulation.
>
> Second, although system-level efficiency was not the primary focus of our submission, we analyzed the inference pipeline and removed redundant operations, improving our reconstruction time to 1.8 s. If the iterative refinement module is disabled, our model becomes Flux4D-base in the main paper, which reconstructs the scene in **0.13 s** and still achieves significant visual quality improvements over existing unsupervised feed-forward methods (DepthSplat, L4GM, STORM) on dynamic actors as shown below. These results are obtained using a single RTX A5000 GPU. Running on a more advanced GPU such as an H100 would also provide a substantial speedup. Additionally, inference acceleration via TensorRT or quantization is straightforward and expected to yield further gains.
>
> | Method             | PSNR_D↑ | SSIM_D↑ | Recon Speed↓       |
> |--------------------|---------|---------|--------------------|
> | DepthSplat         | 16.87   | 0.425   | 0.87s     |
> | L4GM               | 17.36   | 0.343   | 0.32s     |
> | STORM              | 17.65   | 0.367   | **0.07s** |
> | Flux4D-base        |**18.89**|**0.472**| 0.13s     |
> |--------------------|---------|---------|-----------|
> | Flux4D (reference) |  21.99  | 0.662   | 1.8s      |
>
> **Legend:** _D = Dynamics only
>
> Third, while some existing feed-forward methods offer faster inference, they typically operate at significantly lower resolutions (e.g., $160 \times 240$ for STORM) and/or suffer from limited fidelity with noticeable artifacts (see Fig. 3). This compromises downstream usability and leads to a larger domain gap. Moreover, these approaches either do not support key simulation capabilities (e.g., actor removal, insertion) or produce significant artifacts, due to insufficient reconstruction quality and scene decomposition.
>
> In conclusion, we reiterate that Flux4D is designed for offline driving-scene simulation. Unlike traditional pipelines that require manual annotations and hours of optimization, Flux4D offers an order-of-magnitude speedup with competitive fidelity in a fully unsupervised manner, making it well-suited for scalable, simulation-driven autonomy development. We will include updated timing results and discussion in the revision.
>
> [1] Li et al., GVFi: Learning 3D Gaussian Velocity Fields from Dynamic 360 Videos, 2025.\
> [2] Yang et al., STORM: Spatio‑Temporal Reconstruction Model for Large‑scale Outdoor Scenes, 2025.\
> [3] Li et al., Neural Scene Flow Prior, 2021. \
> [4] Li et al., Fast Neural Scene Flow, 2023. \
> [5] Chodosh et al., Re-Evaluating LiDAR Scene Flow for Autonomous Driving, 2024. \
> [6] Khatri et al., I Can't Believe It's Not Scene Flow!, 2024.

---

> > ### Comment · Reviewer_Y9KP · 2025-08-06
> > **Official Comment by Reviewer Y9KP**
> >
> > Thank you to the authors for the detailed response. The rebuttal has effectively addressed my questions, and I have decided to raise my final score to "Borderline accept".

---

> > > ### Author Response · Authors · 2025-08-06
> > >
> > > Thank you again for the thoughtful review and the prompt follow-up discussion. We are glad to hear that our responses helped clarify your concerns. We appreciate your updated evaluation and consideration!
> > >
> > > Please let us know if there are any remaining questions or suggestions, we would be happy to address them.

---

### Official Review · Reviewer_tD1B · 2025-07-02

**Clarity:** 3
**Significance:** 3
**Originality:** 3
**Rating:** 5
**Confidence:** 4

**Summary:**

This paper presents a feed-forward method based on 3D U-Net to predict dynamic 3D Gaussian splats from video frames and LiDAR data. The training process is supervised by reconstruction terms of RGB and depth, along with regularization on the velocity of each Gaussian splat. The predicted properties of splats are then iteratively refined by a network that takes in the current properties and backpropagated gradients to achieve higher visual fidelity. Although being surprisingly concise, this method exhibits clear superiority over several state-of-the-art methods, including those relying on labels for motion. As claimed by the authors, the key ingredient for Flex4D's performance is learning from a diverse range of scenes.

**Questions:**

- What is the consideration of not reporting quantitative comparison with DrivingRecon and STORM is not included?

**Ethical Concerns:**

["NO or VERY MINOR ethics concerns only"]

**Final Justification:**

My questions have been well answered by the authors with sufficient experimental evidence. Therefore, I am glad to still vote for an acceptance.

**Limitations:**

- The authors made an abundant discussion on the limitations in Section F.2. The artifacts shown in Figure A5 are reasonable and interesting.

**Quality:**

4

**Strengths And Weaknesses:**

# Strengths
- This method is simple and effective, achieving faster inference on higher-resolution input while obtaining better results. Being able to predict flow without direct supervision is impressive. In particular, this method outperforms baselines that take motion labels.
- The authors conduct thorough experiments to support the effectiveness of the method. The ablation study shown in Table 5 indicates a clear advantage of iterative refinement.
- It is nice to have Section E.2 for a conceptual comparison with DrivingRecon and STORM.

# Weaknesses
- Besides the comparative discussion in Section E.2, Flux4D uses 3D U-Net with sparse convolution to directly process 3D Gaussian splats in the space, which is fundamentally different from DrivingRecon and STORM, which predict pixel-aligned 3DGS from images. How much performance difference is brought by this architecture change? According to Figure A4, Flex4D produces better visual details. But does this improvement in details mainly come from the iterative refinement?

---

> ### Author Rebuttal · Authors · 2025-07-30
>
> Thank you for the thoughtful review and valuable feedback. We are pleased that Flux4D was recognized as “simple and effective,” delivering “faster inference” with “better results,” and that its ability to predict flow without supervision was found “impressive”. As suggested, we have added quantitative comparisons with STORM, highlighted our ablation to isolate the impact of the iterative refinement, and will include additional results and analysis in the revision.
>
> **Q1: Quantitative comparison with DrivingRecon and STORM**\
> **A1:** Thank you for pointing this out. We have conducted additional experiments to compare with STORM. The comparisons with DrivingRecon and STORM are summarized as follows.
>
> ***Comparison to DrivingRecon:***
> We report quantitative comparison with DrivingRecon on WOD following their setting (Line 213–215) in Table 3 (main paper) and provide visual comparisons (Supp Fig. A4). We observe significant improvements by Flux4D (**+5.99 dB in PSNR and +0.21 in SSIM**), demonstrating the effectiveness for unsupervised dynamic scene reconstruction.
>
> ***Comparison to STORM:***
> At the time of submission (May 15), STORM was not open-sourced. Following its release afterwards (May 19), we retrained STORM under the same settings as used in Table 1 of the main paper. Additionally, as suggested by Reviewer QsPn, we also include some representative scene flow metrics, namely Threeway-EPE (EPE-3way) [1] and Bucketed Normalized EPE (Mean S, Mean D) [2]. We observe significant improvements of our method in reconstruction realism and flow estimation. Specifically, Flux4D is able to capture high-speed moving actors (e.g., oncoming actors) and small objects / far-away objects thanks to our architectural design to learn flow and appearance directly in 3D space with an approximate geometry initialization (reduce ambiguity and handle full-resolution $1080 \times 1920$ inputs). Please see Q2/A2 for more discussions. We will provide more visualizations and analysis in the revision.
>
> | Method |PSNR_D↑|SSIM_D↑|D-RSME_D↓|V-RMSE_D↓|PSNR↑|SSIM↑|D-RSME↓|V-RMSE↓|EPE-3way↓|Mean S↓|Mean D↓|
> |----|----|----|----|----|----|----|----|----|----|----|----|
> | STORM  | 17.65 | 0.367| 5.24| 0.203| 20.79| 0.508| 4.80 | 0.238 | 0.201 | 0.076|0.931|
> | Flux4D |**21.99**|**0.662**|**1.63**|**0.157**|**23.84**|**0.675**|**1.07**|**0.182** |**0.165** |**0.031**|**0.800**|
>
> **Legend:** _D = Dynamics only
>
> **Q2: Directly processing Gaussians in 3D space vs predicting pixel-aligned Gaussians in DrivingRecon and STORM**\
> **A2:** While common in recent feed-forward reconstruction methods [3,4], predicting pixel-aligned Gaussians introduces substantial memory and computational overheads, especially in large-scale driving scenes. This design forces methods such as STORM and DrivingRecon to significantly downsample the image (e.g., ×8 in STORM and ×5 in DrivingRecon), limiting reconstruction fidelity and scalability. In contrast, Flux4D is designed to support high-fidelity, full-resolution ($1920 \times 1080$) reconstruction and simulation. To this end, it adopts a fundamentally different architecture that learns flow and appearance directly in 3D space with an approximate geometry initialization. This 3D design yields a more compact and geometrically consistent representation across views, improving scalability and enabling explicit multi-view flow reasoning. It also reduces appearance-motion ambiguity.
>
> As STORM and DrivingRecon do not rely on LiDAR at test time (they used LiDAR as supervision during training same as Flux4D), we also conduct additional experiments to demonstrate that Flux4D can also operate in a LiDAR-free mode (while still processing Gaussians in 3D space) by using off-the-shelf monocular depth estimation model, namely Metrics3D [5]. Specifically, we removed LiDAR points entirely and instead voxelized the projected monocular depth with a 5 cm resolution, retaining up to 2M points, and then trained the model following the same 3D processing pipeline.
>
> We observed that the flow estimation performance remains comparable, and in some cases, the visual realism improves in background regions (e.g., buildings) due to the broader coverage provided by monocular depth, particularly in areas where LiDAR sparsity limits reconstruction quality. Moreover, combining both LiDAR and Metrics3D depth yields the best overall realism.
>
> | Method      |PSNR_D↑|SSIM_D↑|D-RSME_D↓|V-RMSE_D↓|PSNR↑|SSIM↑|D-RSME↓|V-RMSE↓|EPE-3way↓|Mean S↓|Mean D↓|
> |-------------|--------|--------|-----------|------------|------|------|-------|-------|------|-------|-------|
> |STORM        | 17.65  | 0.367  | 5.24      | 0.203      | 20.79| 0.508| 4.80  | 0.238 |0.201 | 0.076|0.931|
> |Flux4D (monocular depth only)  | 21.71  | 0.668  | **1.45**      | 0.159      | 23.87| 0.688| 1.23  | 0.186 |0.165|0.037|0.863
> |Flux4D (lidar, main paper)      | **21.99**  | 0.662  | 1.63      | **0.157**      | 23.84| 0.675| **1.07**  | **0.182** | 0.165|**0.031**|**0.800**|
> |Flux4D (lidar + monocular depth) | **21.99**  | **0.682**  | 1.52      | 0.158      | **24.55**| **0.726**| 1.11  | 0.184 | **0.161**|0.032|0.873
>
>
> **Legend:** _D = Dynamics only
>
> **Q3: Improvement from Iterative Refinement**\
> **A3:** The iterative refinement module in Flux4D is designed primarily to enhance reconstruction and simulation fidelity and aim to bridge the gap with state-of-the-art per-scene optimization approaches like StreetGS and NeuRAD. It helps improve fine-grained visual details and reduce artifacts as shown in Tab. 5 and Fig. A2. However, the core motion reasoning and flow estimation are already well captured by the single forward-pass model, which is named Flux4D-base in the Main paper.
>
> To illustrate this, we summarize the metrics for STORM, Flux4D-base, and the full Flux4D model below. As shown in the table, Flux4D-base already outperforms STORM significantly across all metrics, especially in visual quality of dynamic regions and flow estimation, while the full model with refinement further improves overall reconstruction fidelity (PSNR, SSIM).
>
> | Method      |PSNR_D↑|SSIM_D↑|D-RSME_D↓|V-RMSE_D↓|PSNR↑|SSIM↑|D-RSME↓|V-RMSE↓|EPE-3way↓|Mean S↓|Mean D↓|
> |----|----|----|----|----|----|----|----|----|----|----|----|
> | STORM  | 17.65 | 0.367| 5.24| 0.203| 20.79| 0.508| 4.80 | 0.238 | 0.201 | 0.076|0.931|
> | Flux4D-base | 18.89  | 0.472  | 1.98  | 0.165 | 20.91| 0.583| 1.34 | 0.193 |**0.164**|**0.030**|0.835
> | Flux4D |**21.99**|**0.662**|**1.63**|**0.157**|**23.84**|**0.675**|**1.07**|**0.182** |0.165|0.031|**0.800**|
>
> **Legend:** D = Dynamics only
>
> [1] Chodosh et al., Re-Evaluating LiDAR Scene Flow for Autonomous Driving, 2024. \
> [2] Khatri et al., I Can't Believe It's Not Scene Flow!, 2024. \
> [3] Charatan et al., pixelSplat: 3D Gaussian Splats from Image Pairs for Scalable Generalizable 3D Reconstruction, 2024. \
> [4] Xu et al., DepthSplat: Connecting Gaussian Splatting and Depth, 2025. \
> [5] Hu et al., Metric3d v2: A versatile monocular geometric foundation model for zero-shot metric depth and surface normal estimation, 2024.

---

> ### Author Response · Authors · 2025-08-07
> **Looking forward to follow-up discussions!**
>
> We thank the reviewer for taking precious time in checking our responses. We hope our answers and additional results address your concerns well. Specifically,
>
> **Q1/A1**: We compared DrivingRecon in Table 3 (main paper) and conducted additional comparisons with STORM as suggested. \
> **Q2/A2**: We clarified that Flux4D adopts a fundamentally different architecture that learns flow and appearance directly in 3D space with an approximate geometry initialization. This 3D design yields a more compact and geometrically consistent representation across views, improving scalability and enabling explicit multi-view flow reasoning. It also reduces appearance-motion ambiguity. \
> **Q3/A3**: We highlighted the benefits of the iterative refinement module, which further improves reconstruction fidelity beyond the already strong performance of the single forward-pass model (Flux4D-base).
>
> Thanks again for the insightful review and suggestions. Please let us know if you have any additional or follow-up questions. We will be more than happy to address them. Any follow-up discussions are highly appreciated!

---

> > ### Comment · Reviewer_tD1B · 2025-08-07
> >
> > I thank the authors for their detailed response and the extensive experimental results. It is clear that the iterative refinement significantly improves photometric accuracy, and that the architectural changes are beneficial for scene flow learning. I am glad to still vote for an acceptance.

---

> > > ### Author Response · Authors · 2025-08-07
> > > **Thanks for the prompt reply!**
> > >
> > > Thank you for the thoughtful suggestions and follow-up discussion. We are glad to hear that our experimental results clarify the impact of our architectural design and iterative refinement clearly.
> > >
> > > Thanks again for the insightful review and suggestions! Please let us know if you have any further comments or questions.

---

### Official Review · Reviewer_QsPn · 2025-07-06

**Clarity:** 4
**Significance:** 4
**Originality:** 3
**Rating:** 5
**Confidence:** 4

**Summary:**

Flux4D is a method for 4D reconstruction using data driven priors and a (mostly) feed-forward architecture that allows for fast inference without requiring expensive per-scene optimization. The method uses RGB + LiDAR inputs and uses gaussians to render novel-views. The gaussians can be transformed in 3D space using learned velocity offsets (flow) and the newly rendered views are used for supervision with a reconstruction loss. This technique allows for a 3D sparse convolutional U-Net to learn the temporal dynamics of the scene in a fully self-supervised way.

The authors demonstrate the performance of this 4D scene reconstruction technique on a variety of tasks such as scene editing, static/dynamic decomposition, flow prediction and more. The method achieves SOTA performance on reconstruction metrics compared to other unsupervised methods such as EmerNeRF and DeSiRe-GS.

**Questions:**

1. Could you evaluate scene flow performance on a common benchmark such as Argoverse 2 or WOD and either submit to the leaderboard or provide a common metric to support the scene flow performance claims? The `bucketed-scene-flow-eval` python package ([bucketed-scene-flow-eval·PyPI](https://pypi.org/project/bucketed-scene-flow-eval/)) provides implementations for both 3-way EPE and bucket normalized EPE with support for both Argoverse 2 and WOD. Some numbers here (even if not SOTA!) would easily raise my score to accept.
2. Could you please add some 3D visualizations in the appendix (or better an interactive website) so we can view the point clouds/gaussians of the generated novel-views? Similar to those shown in Figure 2.
3. In the motion enhancement section, it is stated that flows are rendered to the image plane and used to upweight the photometric loss. Is this using the flow predictions? What happens at the start of training when presumably the velocity estimates are not good?
4. In the paper it is mentioned that you "support" LiDAR input which seems to imply that Flux4D could work without it, (though the limitations state that it's necessary). Is it strictly necessary? Have you tried an experiment with monocular depth estimation only?

**Ethical Concerns:**

["NO or VERY MINOR ethics concerns only"]

**Final Justification:**

The authors addressed my concerns about the limited evaluations for their auxiliary claims around scene flow. The updated metrics show that while the new method provides improvements in flow estimation compared to prior reconstruction methods, it still falls short of dedicated scene flow methods. This contextualization provides important context on real world limitations as well as an exciting direction for future work. As a result, I raise my score to accept.

**Limitations:**

yes

**Quality:**

3

**Strengths And Weaknesses:**

## Strengths
The strengths of this paper are the simple construction of the method and the ability for it to scale with data. The qualitative examples provided, and quantitative metrics support the impressive results, and the baselines are thoughtfully chosen.

## Weaknesses
The main weakness in my eyes is the metrics used and the lack of 3D results (either quantitative or qualitative). The authors have not provided any of the commonly used scene flow metrics (at a minimum 3-way EPE, but ideally the newer Bucket Normalized EPE) to support the flow performance. In addition no 3D visualizations are provided, and all metrics are on rendered 2D images or depth maps. These evaluations (while common in the relevant literature) often mask the true performance of reconstruction methods.

---

> ### Author Rebuttal · Authors · 2025-07-30
>
> Thank you for the thoughtful review and valuable feedback. We are pleased that the method was recognized for its “simple” design, “ability to scale with data,” and “impressive” results. As suggested, we conducted additional experiments comparing with scene flow methods using standard scene flow metrics, supported interactive 3D visualizations, and evaluated Flux4D without LiDAR (using monocular depth estimation). We will include the results and analysis in the revision.
>
> **Q1: Evaluations on scene flow**\
> **A1:** Thank you for the suggestion. While Flux4D primarily focuses on reconstruction and is not specifically designed for scene flow estimation, we agree that evaluating its performance compared with scene flow estimation methods using standard scene flow metrics is valuable. To this end, we conducted additional experiments comparing Flux4D with NSFP [1] and FastNSF [2] on PandaSet and Waymo Open Dataset (WOD). We also include comparisons with STORM [3].
>
> As existing scene flow estimation methods cannot directly predict flows at novel timesteps, we evaluate scene flow on the input frames, following STORM. We restrict evaluation to LiDAR points within the camera field of view (FoV) and report standard metrics including EPE3D, Acc@5, Acc@10, angular error ($\theta$ error), Threeway EPE (BS: Background-Static, FS: Foreground-Static, FD: Foreground-Dynamics) [4], bucketed normalized EPE [5], and inference speed. For bucketed normalized EPE on WOD, as the provided semantic labels are not rich, we follow EulerFlow [6] and report the metrics on Background (with Signs), Vehicles, Pedestrian, and Cyclists only.
>
> Although not specifically designed for scene flow estimation, Flux4D achieves superior performance across most scene flow metrics using only reconstruction-based supervision (RGB + depth). Notably, it outperforms other methods on smaller or less common object categories such as wheeled VRUs, other vehicles, and pedestrians, as shown in bucketed evaluations. We will include these results, visualizations, and further analysis in the revised version.
>
> ***Pandaset Table (1/2)***
> | Method |EPE3D↓|Acc@5↑|Acc@10↑|$\theta$ error↓|EPE-BS↓|EPE-FS↓|EPE-FD↓|EPE-3way↓|Inference↓|
> |--------|------|------|-------|---------|-------|-------|------|--------|------------|
> | NSFP   |0.183 |0.558 |0.713  |0.510    |0.106 |0.103 |0.573 |0.227   |~5.57 s/frame |
> | FastNSF|0.194 |0.571 |0.714  |0.471    |0.155 |0.134 |0.428 |0.211   |~0.68 s/frame |
> | STORM  |0.120 |0.757 |0.782  |0.489    |**0.009**|**0.098**|0.536 |0.201   |**~0.01 s/frame**|
> | Flux4D |**0.094**|**0.775**|**0.807**|**0.123**|0.019 |0.117 |**0.391**|**0.165**|~0.31 s/frame |
>
> ***Pandaset Table (2/2)***
> | Method |BG-S↓|CAR-S↓|CAR-D↓|WVRU-S↓|WVRU-D↓|VEH-S↓|VEH-D↓|PED-S↓|PED-D↓|Mean S↓|Mean D↓|
> |--------|-----|-----|-----|-----|-----|-----|-----|-----|-----|-----|-----|
> | NSFP   |0.128|0.093|0.668|0.046|0.975|0.060|0.819|0.071|0.945|0.080|0.852|
> | FastNSF|0.196|0.153|**0.581**|0.043|0.960|0.075|0.701|0.041|**0.894**|0.102|**0.784**|
> | STORM  |**0.005**|0.087|0.713|**0.000**|1.000|0.195|1.000|0.093|1.012|0.076|0.931|
> | Flux4D |0.019|**0.078**|0.701|0.011|**0.866**|**0.021**|**0.661**|**0.027**|0.966|**0.031**|0.800|
>
> **Legend:**  BG = Background, CAR = Car, WVRU = Wheeled VRU, VEH = Other Vehicles, PED = Pedestrian, S = Static, D = Dynamics
>
> ***WOD Table (1/2)***
> | Method |EPE3D↓|Acc@5↑|Acc@10↑|$\theta$ error↓|EPE-BS↓|EPE-FS↓|EPE-FD↓|EPE-3way↓|Inference↓|
> |------|------|------|----------|-------|-------|-------|---------|---------|----------|
> | FastNSF|0.162|0.734|0.805|0.908|0.131|0.076|0.650|0.203|~0.28 s/frame|
> | Flux4D |**0.048**|**0.901**|**0.929**|**0.540**|**0.011**|**0.012**|**0.440**|**0.114**|**~0.20 s/frame**|
>
> ***WOD Table (2/2)***
> | Method |BG-S↓|VEH-S↓|VEH-D↓|PED-S↓|PED-D↓|CYC-S↓|CYC-D↓|Mean S↓|Mean D↓|
> |--------|----------|-----|-----|-----|-----|-----|-----|------|------|
> | FastNSF|0.144|0.043|0.653|0.044|1.026|**0.026**|0.744|0.064|0.807|
> | Flux4D |**0.011**|**0.009**|**0.502**|**0.014**|**0.680**|**0.026**|**0.732**|**0.015**|**0.638**|
>
> **Legend:**  BG = Background, VEH = Vehicles, PED = Pedestrian, CYC = Cyclists, S = Static, D = Dynamics
>
> **Q2: 3D visualizations in the appendix (or better an interactive website)?** \
> **A2:** Thank you for the suggestion. We now support interactive 3D visualizations, including 3D point, camera and flow visualization using the `viser` library, as recommended. Due to NeurIPS rebuttal guidelines prohibiting any links, we are unable to share the interactive visualizations at this stage. However, we will include representative 3D visualizations in the camera-ready version and provide interactive examples on the project website.
>
> **Q3: Velocity reweighting: Is this using the flow predictions? What happens at the start of training when presumably the velocity estimates are not good?** \
> **A3:** Yes, we use the predicted flow to reweight the photometric loss and enhance the supervision for dynamic regions. As detailed in Supplementary Sec. A.1 (Eq. 1, Line 16-20), we apply a pixel-wise reweighting based on the rendered velocity magnitude:
>
> $$
> \mathcal{L} = (1 + \left\Vert \text{sg}(\mathbf{v_r}) \right\Vert) \cdot \mathcal{L}_{\text{rgb}},
> $$
>
> where $\text{sg}(\cdot)$ denotes the stop-gradient operation and $\mathbf{v_r}$ is the rendered velocity in image space. For stability, we clip $\left\Vert\mathbf{v_r}\right\Vert$ to $[0, 10]$.
> At the start of training, the predicted velocity magnitudes are close to zero, so all pixels are weighted equally. Even when early velocity estimates are inaccurate, the clipping and base weighting ensure stable gradients and effective learning throughout training.
>
>
> **Q4: Flux4D without LiDAR (e.g., monocular depth estimation)** \
> **A4:**  Thank you for pointing this out. Although LiDAR is an important component of Flux4D, it is not strictly necessary. In our framework, LiDAR primarily serves as an initialization for the 3D scene structure and does not need to be highly accurate. To evaluate performance without LiDAR, we conducted an additional experiment using monocular depth estimated by off-the-shelf Metric3D [7] model, as suggested. Specifically, we removed LiDAR points entirely and instead voxelized the projected monocular depth with a 5 cm resolution, retaining up to 2M points, and then trained the model following the same 3D processing pipeline. The results are shown below.
>
> We observed that the flow estimation performance remains comparable, and in some cases, the visual realism improves in background regions (e.g., buildings) due to the broader coverage provided by monocular depth, particularly in areas where LiDAR sparsity limits reconstruction quality. Moreover, combining both LiDAR and Metrics3D depth yields the best overall realism.
>
> We will include the quantitative results, visualizations, and further analysis in the revision.
>
> | Method      |PSNR_D↑|SSIM_D↑|D-RSME_D↓|V-RMSE_D↓|PSNR↑|SSIM↑|D-RSME↓|V-RMSE↓|EPE-3way↓|Mean S↓|Mean D↓|
> |-------------|--------|--------|-----------|------------|------|------|-------|-------|------|-------|-------|
> |STORM        | 17.65  | 0.367  | 5.24      | 0.203      | 20.79| 0.508| 4.80  | 0.238 |0.201 | 0.076|0.931|
> |Flux4D (monocular depth only)  | 21.71  | 0.668  | **1.45**      | 0.159      | 23.87| 0.688| 1.23  | 0.186 |0.165|0.037|0.863
> |Flux4D (lidar, main paper)      | **21.99**  | 0.662  | 1.63      | **0.157**      | 23.84| 0.675| **1.07**  | **0.182** | 0.165|**0.031**|**0.800**|
> |Flux4D (lidar + monocular depth) | **21.99**  | **0.682**  | 1.52      | 0.158      | **24.55**| **0.726**| 1.11  | 0.184 | **0.161**|0.032|0.873
>
>
> **Legend:** _D = Dynamics only
>
> [1] Li et al., Neural Scene Flow Prior, 2021. \
> [2] Li et al., Fast Neural Scene Flow, 2023. \
> [3] Yang et al., STORM: Spatio-Temporal Reconstruction Model for Large-Scale Outdoor Scenes, 2025.\
> [4] Chodosh et al., Re-Evaluating LiDAR Scene Flow for Autonomous Driving, 2024. \
> [5] Khatri et al., I Can't Believe It's Not Scene Flow!, 2024. \
> [6] Vedder et al., Neural Eulerian Scene Flow Fields, 2025.  \
> [7] Hu et al., Metric3d v2: A versatile monocular geometric foundation model for zero-shot metric depth and surface normal estimation, 2024.

---

> > ### Comment · Reviewer_QsPn · 2025-08-05
> >
> > Thanks to the authors for the thoughtful rebuttal, the additional evaluations help to contextualize the results quite nicely! The flow performance seems to be generally improved compared to other reconstruction methods (ie. STORM), and can be on-par with FastNSF in some categories, but still falls short of the SOTA dedicated scene flow methods. This lines up with my intuition & experience with these kinds of methods, and provides an interesting direction for future work that might be able to unify SOTA flow with SOTA reconstruction.
> >
> > I also appreciate the clarification on the pixel-wise reweighing and the monocular depth experiment. I raise my score to accept :)

---

> ### Author Response · Authors · 2025-08-05
> **Thanks for the prompt reply!**
>
> Thank you for the thoughtful suggestions and follow-up discussion. We are glad to hear that the additional evaluations helped contextualize the results. We agree that unifying SOTA flow and SOTA reconstruction is an exciting direction for future work, with the potential to benefit both tasks and enhance generalization. We hope that Flux4D takes an important step toward building this connection!
>
> Thanks again for the insightful review and suggestions! Please let us know if you have any further comments or questions.

---

### Note · Authors · 2025-08-11

We sincerely thank all reviewers for their valuable feedback and thoughtful discussions.

Flux4D achieves high-fidelity 4D reconstruction and simulation of large-scale driving scenes in a fully unsupervised manner. Reviewers found our approach “simple and effective” [**Reviewer QsPn, Reviewer tD1B**], and acknowledged our evaluation is “sufficient” with “thoughtfully chosen baselines” [**Reviewer Ew2X, Reviewer QsPn**], the results are “impressive” [**Reviewer QsPn, Reviewer tD1B**].

We are glad that our rebuttals and additional experiments were well-received, leading **Reviewers QsPn** and **tD1B** to recommend acceptance and **Reviewer Y9KP** to borderline accept. Their engagement has significantly strengthened our work.

We appreciate the follow-up discussion with **Reviewer Ew2X** regarding the clarity of our experimental framing. While we highlighted the usage of LiDAR in the introduction (Line 67-69), we will further clarify its role to better address this feedback. Specifically, we will restructure the results section to explicitly detail the geometric priors (LiDAR, monocular depth, or both) used for Flux4D and all baselines in every comparison. This change will better highlight that our method’s core strength persists even in a LiDAR-free setting, as demonstrated in our rebuttal. We also note that all baselines (Tab. 1, 2, 4) also have access to LiDAR either in reconstruction or during training.

Thank you once again for your time and consideration. We are confident that the revised paper will be a strong contribution to the NeurIPS community.

---

### Decision · Program_Chairs · 2025-09-17

**Decision:**

Accept (poster)

**Comment:**

All reviewers appreciate the submission and find the solution technically efficient with strong results. The AC recommends accepting the paper.
The AC agrees with reviewer Y9KP that the evaluations are restricted to autonomous driving scenes, while the title and introduction suggest a broader focus on general 4D. The authors should either moderate their claims by explicitly limiting them to autonomous driving scenarios or incorporate more results on general 4D reconstruction during the revision period.